# The Use of Infrared Thermography (IRT) in Burns Depth Assessment: A Diagnostic Accuracy Meta-Analysis

**Aqua Asif** [1] , **Constantinos Poyiatzis** [1] , **Firas J. Raheman** [2] **and Djamila M. Rojoa** [3,*]

1    George Davies Centre, Leicester Medical School, University of Leicester, Lancaster Rd., Leicester LE1 7HA, UK
2    East and North Hertforshire NHS Trust, Coreys Mill Lane, Stevenage SG1 4AB, UK
3    Imperial College NHS Trust, Praed Street, London W2 1NY, UK
*    Correspondence: djamila.rojoa@doctors.org.uk

**Abstract:** Background: The timely diagnosis of burns depth is crucial to avoid unnecessary surgery and delays in adequate management of patients with burn injuries. Whilst it is mostly a clinical diagnosis, indocyanine green, laser Doppler imaging and infrared thermography have been used alongside clinical findings to support the diagnosis. Infrared thermography is a noninvasive technique which uses temperature differences to diagnose tissue burn depth. Our study aims to assess its use in differentiating between superficial and deep burns. Methods: We conducted a systematic literature review and meta-analysis using electronic databases. We used a mixed-effects logistic regression bivariate model to estimate summary sensitivity and specificity and developed hierarchical summary receiver operating characteristic (HSROC) curves. Results: We identified 6 studies reporting a total of 197 burns, of which 92 were proven to be deep burns. The reference standard was clinical assessment at the time of injury and burn healing time. The pooled estimates for sensitivity and specificity were 0.84 (95% CI 0.71–0.92) and 0.76 (95% CI 0.56–0.89), respectively. Conclusions: IRT is a promising burns assessment modality which may allow surgeons to correctly classify burn injuries at the time of presentation. This will allow a more efficient management of burns and timely surgical intervention.

**Keywords:** burns; depth assessment; infrared thermography; diagnosis

## 1. Introduction

Infrared thermography [IRT] as a diagnostic modality uses devices that transpose long-wave infrared light into thermal images capable of displaying high levels of visual contrast between areas with small differences in temperature [1]. Medical applications of IRT are founded in the physiology of cellular metabolism as the body releases heat predictably in healthy skin, but damage to blood vessels results in measurable changes to heat loss via radiation [2]. The reliability of IRT combined with the physiological mechanisms that enable it to identify temperature differences have promising implications on determining the depth of a burn, which has relied primarily on clinical evaluation and the time to healing and reducing the associated risk of hypertrophic scar development. The assessment of burn depth and severity is aimed at answering whether the burn will heal within a few weeks or will require surgical intervention. Burn-depth analysis thus drives the need for surgical intervention or conservative treatment. Burns are categorized as superficial or superficial partial-thickness, which do not require intervention, or deep-partial thickness or full thickness, which do require involvement. Surgery seeks to primarily restore functionality of the tissue and secondarily address cosmesis. Answering this question promptly, accurately, and with a minimally invasive approach is paramount to recovery time and is currently conducted via the employment of several techniques.

Clinical assessment by burn specialists, the mainstay for burn-depth determination, leads to only 60 to 75% accuracy rates of diagnosis [3]. Numerous techniques have been

implemented to address this but have their own limitations. Examples include infrared thermography, laser Doppler imaging (LDI), and indocyanine green (ICG) angiography.

Indocyanine green is a fluorescent dye administered intravenously to measure tissue perfusion. Accordingly, some limitations of indocyanine green angiography include being an invasive procedure, its rapid blood clearance, and the requirement of expensive equipment, reducing its overall widespread use [4,5]. Wounds with intact skin have been misinterpreted as deep burns as a result of some cases reporting melanin being able to absorb similar wavelengths to the ones used in indocyanine green angiography [6,7].

Tissue punch biopsy followed by histological investigation is currently the gold standard for burn-depth assessment. It is subject to interobserver variations between pathologists and is an invasive, timely, and costly procedure to carry out. For large area burns particularly, histological analysis can display heterogeneous sampling errors [8–10].

Diagnostic infrared thermography is noninvasive and has minimal side effects whilst exceeding the accuracy of clinical evaluations and rivalling ICG or LDI techniques [11]. Static infrared imaging ascertains burn depth via spotting temperature differences—full thickness burns are cooler in temperature than healthy skin or superficial burns due to damaged or destroyed blood vessels [12]. Active infrared imaging uses cold excitation prior to observation to measure the return time to normal temperatures [13–15]. Superficial burns will return to normal temperatures faster than full thickness burns. Both active and static imaging can be completed in minutes and are noninvasive procedures with real-time results. The accuracy of both methods exceeds clinical evaluation with active imaging, having an 83% accuracy rate [15]. Infrared cameras are further easy to operate, and although standardized protocols and imaging assessments are required, the objectivity and reproducibility makes IRT an attractive option compared to more invasive diagnostic modalities. The purpose of our study is to investigate the accuracy of IRT in determining burn depth for optimal management.

## 2. Methodology

### 2.1. Design

A study protocol set out the objectives of the review, study inclusion criteria, and methods of analysis. The review was reported in accordance with the Preferred Reporting Items for Systematic Review and Meta-analysis (PRISMA) of diagnostic test accuracy studies [16].

### 2.2. Criteria for Considering Studies for This Review

#### 2.2.1. Type of Studies

Diagnostic studies assessing the use of thermographic imaging in burn-depth determination were analysed. These included both retrospective and prospective cohort studies. Case series, review articles, and conference abstracts and commentaries were not included. Cadaveric and animal studies were also excluded.

#### 2.2.2. Participants

Participants of any age and gender, that have undergone an evaluation of burn depth using thermographic imaging were included. Injuries of varying severity (depth) were analysed, including surface, partial thickness, and full thickness, which were managed surgically or conservatively with follow-up.

#### 2.2.3. Target Condition

Burns range from superficial, superficial partial-thickness, deep partial-thickness, and full thickness. Burns treatment follows a similar gradient from conservative treatments to surgical intervention using skin excision and grafting. Burn severity gradually shifts from healing with time to needing a surgeon's involvement. Quickly determining the proper treatment is time sensitive and functionally the aim of burn-depth analysis. For this study, we classified burns which healed without any surgical intervention as superficial and burns

which required excision as deep. The latter includes deep partial thickness burns requiring surgery and/or full-thickness burns, as described in the included studies.

### 2.2.4. Index Test

Our study aims to analyze the accuracy of infrared thermography (IRT) as a diagnostic tool in determining burn depth. Performing an IRT requires minimal training and can have results in a matter of minutes [17]. IRT provides clinicians with a noninvasive method to determine if surgical intervention is necessary and has an accuracy rate of 83% [15]. The most accurate IRT test is active infrared imaging which works by using cold excitation on both the burn site and healthy skin and measuring the time to return to body temperature [13–15]. The advantage of IRT is determining the separation between superficial partial-thickness and deep partial-thickness burns as deeper burns will return to body temperature slower than superficial and superficial partial thickness burns.

### 2.2.5. Reference Standards

Tissue punch biopsy followed by histological investigation is the current gold standard for burn-depth assessment but can be subject to heterogeneous sampling errors in large area burns and is a costly, invasive procedure. ICG angiography and LDI are two less invasive diagnostic options but require expensive equipment prohibitive for widespread use [4,5]. The most commonly used diagnostic method is clinical assessment by burn specialists, but due to the ambiguity between superficial partial-thickness burns and deep partial-thickness burns, clinical assessment only has a 60 to 75% accuracy rate [3].

### *2.3. Search Methods for Identification of Studies*

The literature search was undertaken over a 2-week period, from 5 August 2020 to 19 August 2020 by three review authors (DMR, AA and CP). The Healthcare Databases Advanced Search (HDAS) interface by the National Institute for Health and Care Excellence (NICE) was used to run searches in the PubMed/Medline/EMBASE/Ovid databases. A list of search strategy keywords was compiled and used to search the literature (Supplementary File S1). There was no language restriction placed on the literature search.

### *2.4. Data Collection and Analysis*
#### 2.4.1. Selection of Studies

Studies were assessed for eligibility by three review authors (DMR, AA, CP) independently. A fourth author (FJR) acted as an arbitrator when consensus could not be reached between the three. The titles and abstracts of the electronic database search results were screened. Full-text articles of the studies meeting the included criteria were then reviewed.

#### 2.4.2. Data Extraction

Data was retrieved by two review authors (AA and CP) and cross-checked by a third and fourth author (DMR and FJR). The following data were extracted from the full-text articles of the selected studies:

- Article (author, year and journal of publication)
- Study design (sample size, type of study)
- Study population and demographics (age, gender)
- Reference standard (clinical assessment and follow-up to assess wound healing)
- Index test (IRT Imaging) and its interpretations
- Quality assessment of the included studies using the Quality Assessment of Diagnostic Accuracy Studies-2 (QUADAS-2) tool
- Data for two-by-two contingency tables (absolute numbers of true positives, false positives, true negatives, false negatives, positive predictive value, negative predictive value, sensitivity, specificity).

## 2.5. Assessment of Methodological Quality

The QUADAS-2 tool on RevMan 5.3 (Cochrane Collaboration, Copenhagen, Denmark) was used to assess the methodological quality of each study. Quality assessment was performed and checked by two authors (CP and AA). The risk of bias was assessed on the four QUADAS-2 domains, "patient selection", "index test", "reference standard", and "flow and timing". We also assessed applicability for "patient selection", "index test", and "reference standard". We answered questions within each domain with "yes", "no", or "unclear", "yes" denoting low risk of bias and "no" denoting a high risk of bias. Based on those answers, the domain was later classified as having "high", "low", or "unclear" risk of bias.

### Statistical Analysis and Data Synthesis

Dichotomous outcomes for both reference standards and index tests were identified from the included studies. Two-by-two contingency tables were constructed after identifying true positives, true negatives, false positives, and false negatives and classified based on burns severity. A mixed-effects logistic regression bivariate meta-analysis model was used for each index test interpretation and logit transformed sensitivities and specificities were modelled. Based on the results from this model, summary receiver operating characteristic curves (SROC) were constructed, and summary sensitivities and specificities were calculated. Hierarchical modelling was used to better appreciate between study heterogeneity, and hierarchical summary ROC (HSROC) were constructed to compute 95% confidence and 95% prediction regions. Based on our diagnostic modelling, the conditional probability of target condition (burn depth) was computed using a probability modifying plot. All analyses were performed using "midas" and "metandi" in Stata 16 (Stata-Corp, College Station, TX, USA). The number of studies was inadequate to assess reporting bias by funnel plot.

## 3. Results

### 3.1. Results of the Search

A total of 1035 reports were identified in our literature search on MEDLINE/EMBASE/ Ovid databases for "thermography" and for "burn depth". Out of these 1035 articles, 55 were selected through title and abstract screening. Some 38 studies were excluded as they were not relevant, and 17 were selected for full article screening based on the inclusion criteria. Next, relevant data extraction could not be conducted for 11 articles, which left 6 studies for inclusion [17–21] in quantitative and qualitative analysis. Figure 1 outlines the literature search process in a PRISMA flow diagram [22].

### 3.2. Characteristics of Included Studies

Table 1 displays the number of patients included in each study and the number of burn sites or regions of interest analysed. The number of types of burns as assessed by the reference standard is shown. Time of imaging post-admission was also noted. A variety of cameras and image analysis software was used in the acquisition of data. We were only able to identify the thermal sensitivity of the device used in one out of six studies [19]. We analysed a total of six studies and a total of 161 patients. In total, we assessed 197 burn wounds or regions of interest. Of those, 92 were classified as superficial burns by the reference standard, and 105 were classified as deep burns. The mean age range was 3 to 40 years with one study exploring only paediatric hand burns [15]. The male to female patient ratio was 3.27. The reference standard of choice for all included studies was clinical outcome [17–19,21,23,24], alongside histological results for one study [21]. The performance time of our index test, IRT, varied with the range being within 24 h to 72 h [17–19,21,23,25].

### 3.3. Methodological Quality of Included Studies

The QUADAS-2 tool was used to assess each study and provide an overall summary, as shown in Figure 2. The main issue presenting in one of the studies [15] revolved around

calculating temperature difference (ΔT) between healthy tissue and burned tissue to classify burn depth by optimizing it to give a maximum specificity instead of calculating at the point where specificity and sensitivity are equal. Moreover, the reference standard used to distinguish between superficial and deep burns was not homogenous across all studies. Ganon et al. [18] measured the ΔT of paediatric burns at three different timepoints T1, T2, T3 (Day 1–3; Day 4–7; Day 8–10, respectively) and calculated the optimal ΔT from ROCs which yielded a 100% specificity by the images obtained at T3. This ensures that deep burns were not misclassified as superficial burns and did not receive the inappropriate treatment such as unnecessary general anaesthesia and perioperative risks of burn excision in paediatric patients. Moreover, blinded assessment was ensured as the interpreter was uninvolved in the clinical decision making. However, they lack inter-observer reliability as the study included only one observer.

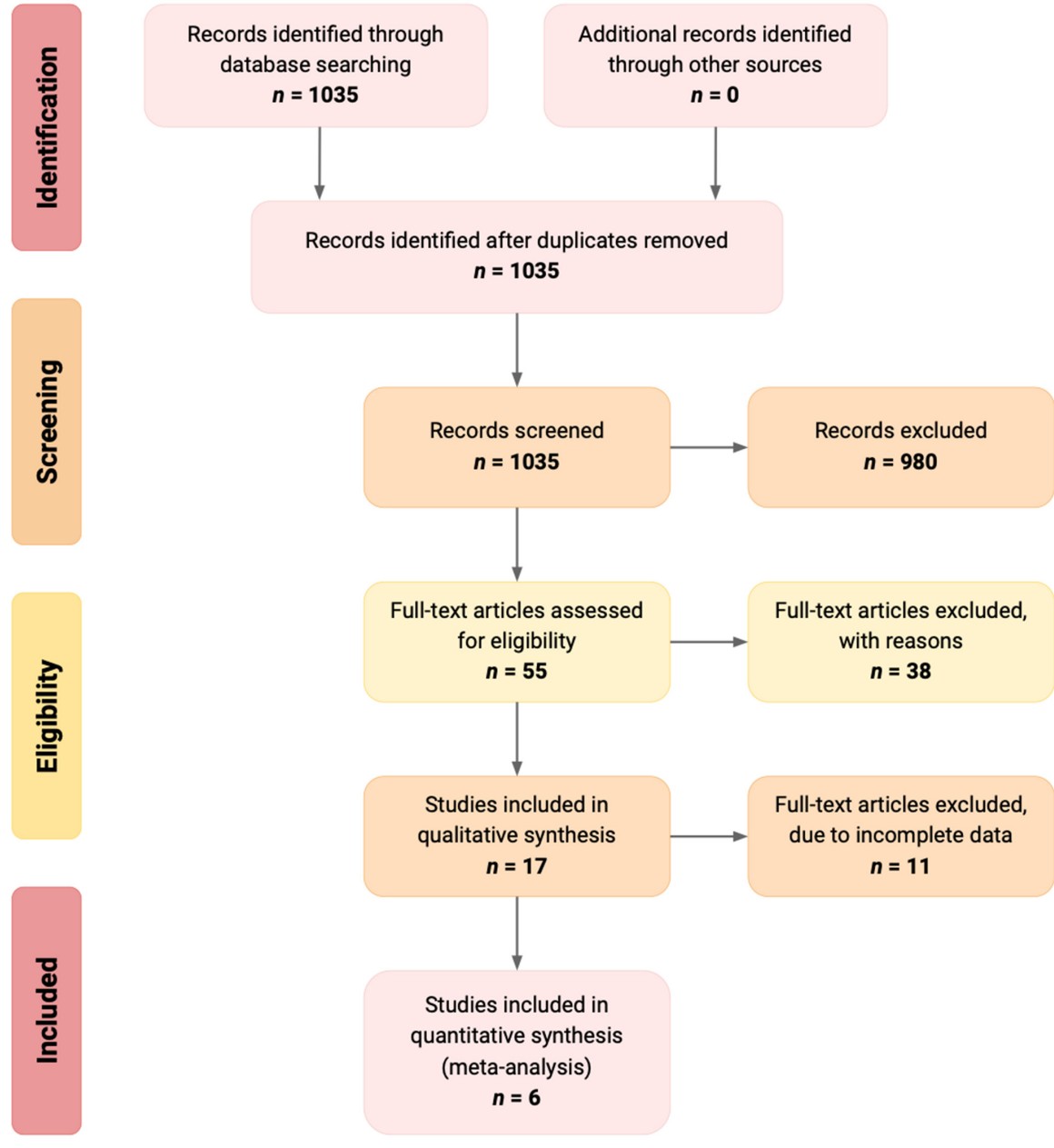

**Figure 1.** Shows the PRISMA diagram for our search strategy.

**Table 1.** Shows the number of patients included in each study and the number of burn sites or regions of interest analysed.

| Study | Study Design | No of Patients | No of Burns or ROI | Age (Range) | M:F Ratio | Type(s) of Burn | | Reference Standard | Point of Measurement | Timing of Clinical Assessment (Days Post Burn) | Timing of Thermograms (Days Post Burn) | Thermogram Thresholds | |
|---|---|---|---|---|---|---|---|---|---|---|---|---|---|
| Ganon et al., 2020 [18] | Prospective cohort | 40 | 40 | 2.9 (13 months–13 years) | 30:10 | Superficial | 29 | Burns healed within day 15 | Exact centre of burned area vs. healthy area > 3 cm away from wound or contralateral distal extremity | Not assessed—outcomes at 15 days used as reference standard | T1 (D1–3) T2 (D4–7) T3 (D8–10) | −1.2 °C diff ones as well | FLIR One, FLIR Systems, Inc., Wilsonville, OR 97070, USA |
| | | | | | | Deep | 11 | Burns not healed (<95% epithelialized wound surface) or have been grafted | | | | | |
| Martinez-Jimenez et al., 2018 [19] | Prospective cohort | 22 | 14 | 26.5 | 16:6 | Superficial | 8 | Re-epithelisation before 15 days | | Not assessed—outcomes at 15 days used as reference standard | Within first 3 days (mean 1.45, median 1) | 3.0 °C (for a specificity of 100%) | FLIR T400, FLIR Systems, Inc., Wilsonville, OR 97070, USA |
| | | | | | | Deep | 6 | Wounds not re-epithelisaed after 15 days (requiring one or more skin grafts or removal of appendage) | | | | | |
| Simmons et al., 2018 [23] | Prospective cohort | 16 | 16 | 37.5 | 11:5 | Superficial | 7 | Wounds not requiring a skin graft | 2 × 2 cm ROI placed on the area of least heat reacquisition within the wound compared to healthy skin | Not assessed—outcomes at 15 days used as reference standard | Day 1 | Temperature difference for healed vs. non-healed wound = −16.8 vs. −23.6 (AUC 1) | Model H, Thermapp, Opgal Optronic Industries Ltd., Karmiel 20101, Israel |
| | | | | | | Deep | 9 | Wound requiring a skin graft | | | | | |
| Wearn et al., 2018 [17] | Prospective cohort | 16 | 52 | 37.5 | 13:3 | Superficial | 9 | LDI and clinical assessment of wounds with a <21 days healing potential | Burn wound temperature based on 52 ROIs compared to control area of non-burnt skin on contralateral side | Not reported—outcomes at 21 days and LDI used as reference standard | Day 0 Day 3 | 1.5 °C | FLIR SC660, FLIR Systems, Inc., Wilsonville, OR 97070, USA |
| | | | | | | Deep | 43 | LDI and clinical assessment of wounds with a >21 days healing potential | | | | | |
| Singer et al., 2016 [21] | Prospective cohort | 24 | 39 | 39.5 (SD 16.4) | 19:5 | Superficial | 23 | Wounds with healing time < 21 days | ROI was a single point in the middle of the burn that appeared to be deepest | Not reported—outcomes at 21 days and wound requiring excision with histological assessment used as reference standard | Within 2 days | 0.1 °C | FLIR T300, FLIR Systems, Inc., Wilsonville, OR 97070, USA |
| | | | | | | Deep | 16 | Wounds with healing time > 21 days, or requiring excision and grafting with histological assessment | | | | | |

**Table 1.** *Cont.*

| Study | Study Design | No of Patients | No of Burns or ROI | Age (Range) | M:F Ratio | Type(s) of Burn | | Reference Standard | Point of Measurement | Timing of Clinical Assessment (Days Post Burn) | Timing of Thermograms (Days Post Burn) | Thermogram Thresholds | |
|---|---|---|---|---|---|---|---|---|---|---|---|---|---|
| Cole et al., 1990 [24] | Prospective cohort | 23 | 36 | 32 | 19:4 | Superficial | 16 | Wound healing < 21 days | Temperature of burned surface (divided into zones) wrapped in clingfilm was used | Not reported—outcomes at 21 days used as reference standard | Within 2 days | Cut-off level of 31 °C to distinguish between warm and cold | AGA Thermovision 782 |
| | | | | | | Deep | 20 | Wound healing > 21 days requiring excision and grafting | | | | | |

Key: ROI—region of interest, M:F—Male to Female ratio, IRT—infrared thermography.

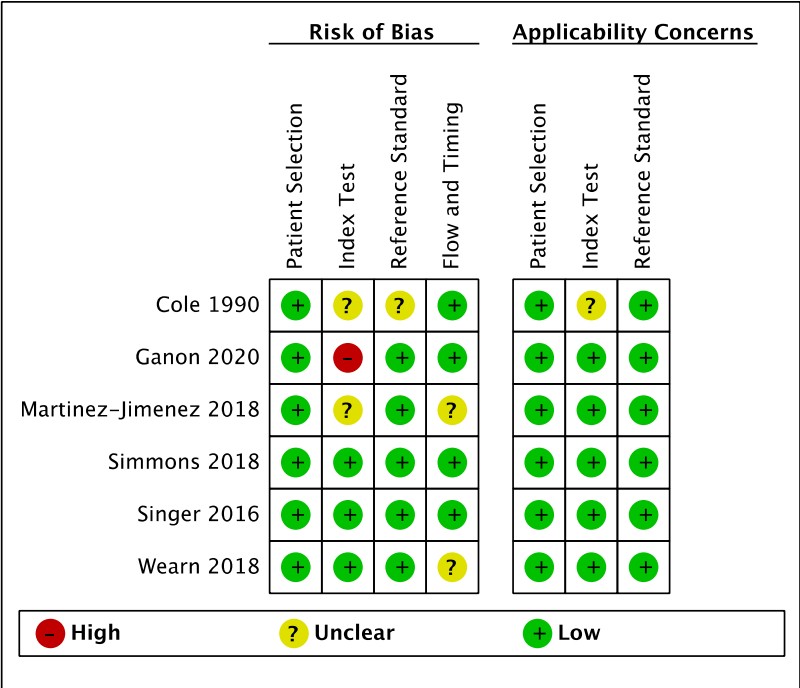

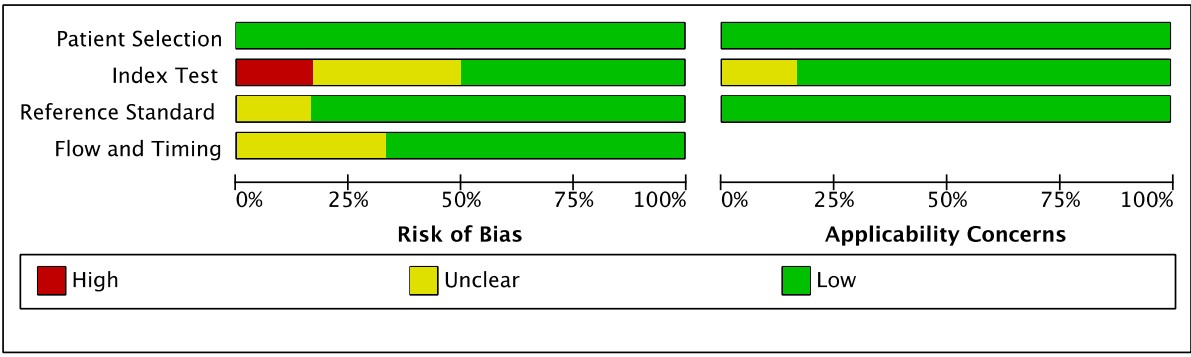

**Figure 2.** Shows the (**A**) methodological quality for each individual study and (**B**) the overall study quality summary.

Of interest was the study by Martínez-Jiménez et al. [19] whereby a development cohort was used to calculate an optimal ΔT from ROCs which was then applied in a treatment cohort to validate the predictive model. Although two different study groups were used to minimize reporting bias, they did not differentiate between paediatric and adult burns in their predictive model. Simmons et al. [23] stipulated that burn surgeons are not able to accurately classify burn depth within 24 h of burn occurrence and require three to five days post-burn to assess for signs of healing for more accurate diagnosis. Their protocol then sought to identify tissue reperfusion kinetics after exposure to a cold challenge in an effort to increase diagnostic accuracy within 24-h post-burn and avoid burn depth progression and unnecessary hospital stays [23]. They have shown a higher sensitivity and specificity (86% and 78%, respectively) compared to what was expected by burn surgeons at 24-h post-burn. However, their sample size was not large enough to clarify the overlap of rewarming kinetics observed in heterogeneous wounds (wound with both deep and superficial burned areas) or further characterize ΔT and features rewarming kinetics of superficial and deep wounds. In Cole et al. [25] despite the larger sample size, we were unable to determine if blinding between the researchers who carried out the index

test and reference standard occurred. This could result in overestimation of the accuracy of their results [21].

### 3.4. Findings

We found that sensitivity ranged from 0.56 to 0.92 and specificity from 0.44 to 0.93, as shown in Figure 3. The pooled estimates for sensitivity and specificity were 0.84 (95% CI 0.71–0.92) and 0.76 (95% CI 0.56–0.89), respectively, as shown in Figure 4A,B. Area under the curve (AUC) was 0.87 (95% CI 0.84–0.90) indicating excellent diagnostic accuracy of IRT when identifying deep burns. The DOR (diagnostic odds ratio) was 16.5 (95% CI 4.4–61.1). The LR+ was 3.49 (95% CI 1.70–7.17), and LR− was 0.21 (95% CI 0.1–0.43). Our probability modifying plot shows the relationship between pre- and post-test probability based on the likelihood of a positive or a negative test for deep burns. Figure 5 shows a predictive probability modifying plot, where infrared thermography appears to be more informative for a negative deep burn (LR− 0.21 [95% CI 0.10–0.43]) and based on prevalence heterogeneity, has a pooled NPV of 0.81 (0.71–0.91) which suggests its ability to rule out deep burns at an early stage.

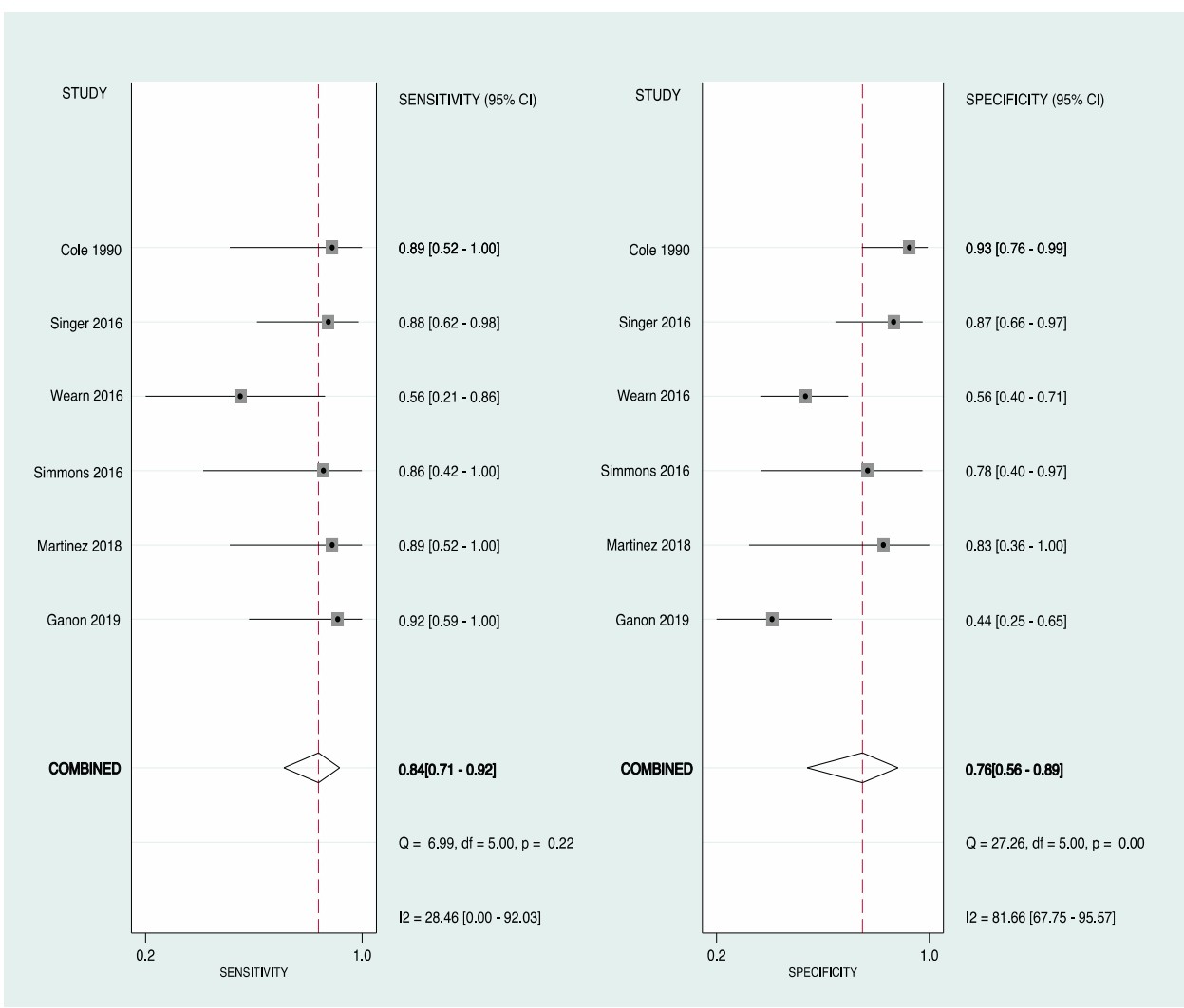

**Figure 3.** Shows sensitivity and specificity forest plots for IRT in the assessment of deep burns.

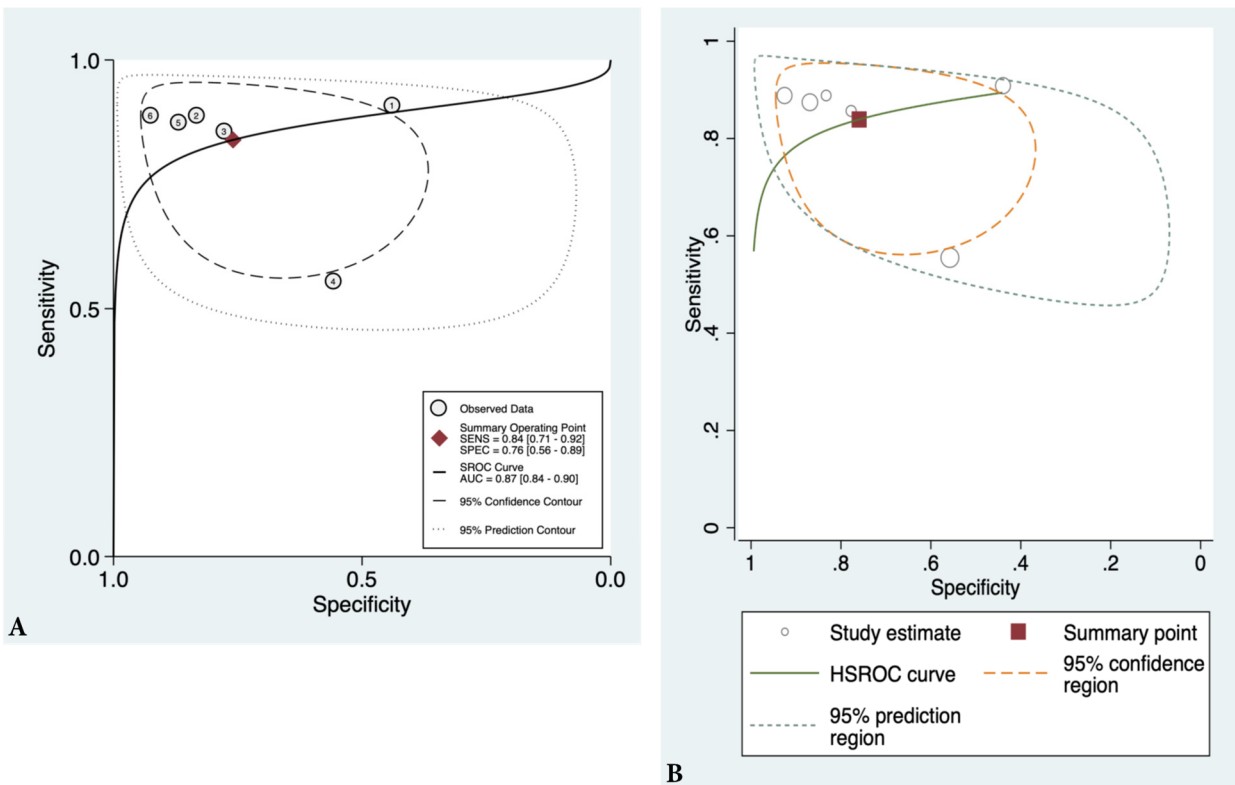

**Figure 4.** Shows (**A**) SROC curve and (**B**) HSROC curve of infrared thermography when assessing burn depth.

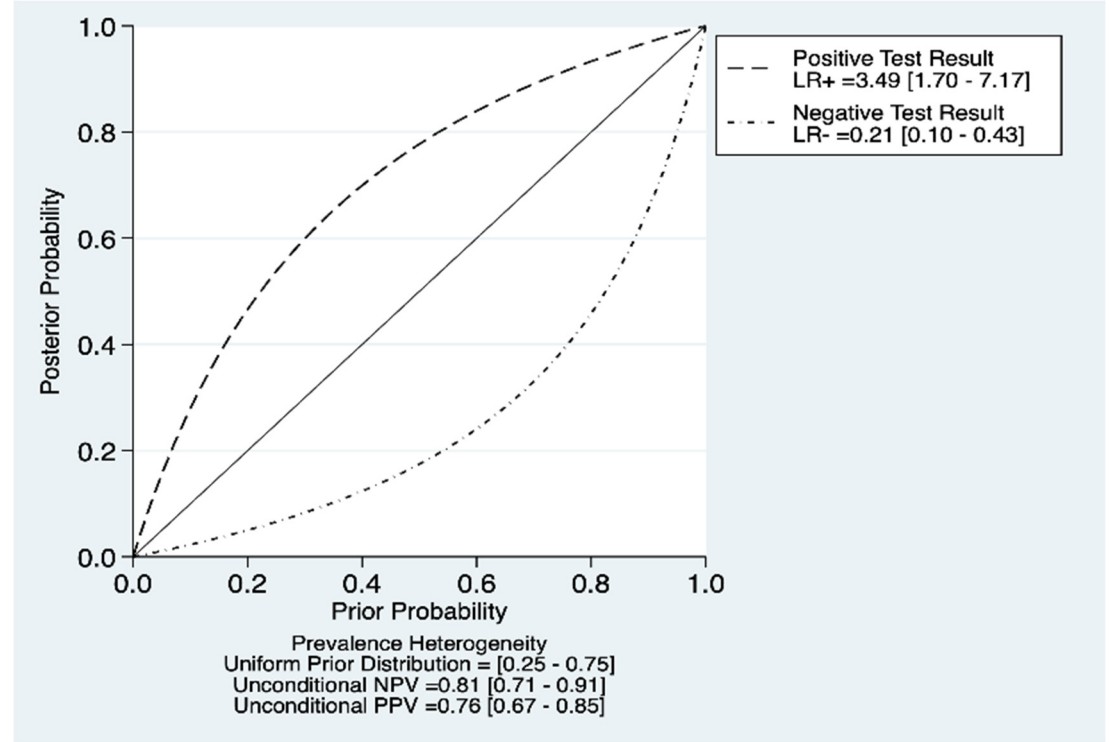

**Figure 5.** Shows the probability modifying plot of infrared thermography when assessing deep burns.

## 4. Discussion

Our meta-analysis is the first to assess the accuracy of infrared thermography in burn-depth analysis. First introduced in 1961, its diagnostic potential has improved exponentially due to technological advancements [25,26]. Together with laser Doppler imaging, infrared thermography is a common objective method used as an adjunct to clinical assessment, enabling the estimation of burn depth and healing time. It is important to have a tool that is cost-effective, accurate, and accessible that will help clinicians to assess burn depth and subsequent treatment options. Deciding between a superficial or a deep burn can have an impact on the treatment and, as such, healing and cosmesis. The current reference tests used to discriminate between superficial and deep burns are riddled with disadvantages, ranging from low accuracy to cumbersome equipment, to invasive procedures. Infrared thermography is a suitable candidate to replace, or act as an adjunct, to clinical assessment—the current, predominant reference—due to its low cost, ease of use, wide availability, and quick interpretation.

In this review, we found six studies [17–19,21,23,25] assessing the diagnostic capability of infrared thermography in assessing burn depth. The pooled sensitivity and specificity were 84% (95% CI 71–92%) and 76% (95% CI 56–89%), respectively. The accuracy of IRT for wound depth assessment has been shown to be greater than clinical assessment only [21] as it detects temperature changes over the first two days. It is thought that this discrepancy in temperature corresponds to wound conversion during the healing of burns. Hence a decrease in temperature may be predictive of a deeper wound. Cole et al. [25] suggested the use of IRT within 48 h of injury is helpful in predicting burns healing within two to three weeks. Therefore, IRT represents an objective burn-depth assessment modality which enables clinicians to determine treatment modality needed by optimizing wound management, reducing unnecessary surgery for superficial burns and minimizing delays in operation for full-thickness burns [19,25]. Furthermore, infrared thermography is a noninvasive contactless technique which minimizes the risk of wound contamination and damage to microcirculation. It is a painless and rapid assessment modality, offering almost instantaneous results and circumventing the need for sedation. Thus, it may be applied to both paediatric and adult patient populations.

Timing of IRT is crucial as the earliest measurement for reliable accuracy has been shown to be 48-h post-burn, after the initial processes have been stabilized, according to Hoeksema et al. [20]. In our analysis, three studies [21,23,25] performed infrared thermography within 48 h, whilst two studies [17,19] performed it within 72 h. Interestingly, Ganon et al. [18] assessed the temperature at three different time frames, T1 (Days 1–3), T2 (Days 4–7) and T3 (Days 8–10), with the highest accuracy found to be at T3. Later scans had a higher accuracy compared to earlier ones. This has also been shown by Wearn et al. [17], where Day 3 scans were more accurate compared to Day 0. However, the diagnostic accuracy of clinical assessment at Day 8 has been shown to be 100% [17]; hence the value of performing thermal assessment lies in the early days post-burn, as it allows early diagnosis and treatment planning during a time period where clinical assessment is inaccurate. Consequently, only T1 findings from Ganon et al. [18] were included in our analysis to minimize between-study heterogeneity. Amongst the included studies, only one [21] used histological results as a reference standard. The remaining ones [17–19,23,25] used the registered healing time instead and the fact that burn wounds underwent excision and grafting. Whilst this might not be considered gold standard, the healing time can be directly correlated to abnormal scarring, which is considered as a key consequence [18]. Moreover, a 15-day healing time cut-off was chosen as scarring is unlikely to occur for healed wounds without grafting within that timeframe [18], whilst after 3 weeks, the risk of hypertrophic scarring is considerably higher.

Whilst temperature change is key to burns wound depth assessment, it is dependent on multiple factors. Martínez-Jiménez et al. [19] have demonstrated that temperature change is significantly affected by age, burn aetiology, depth of injury, and burn area. Hence, having a fixed ΔT might not be the best approach to burns assessment. Moreover, the optimal

cut-off value for differentiating between superficial and deep burns is close to the minimum detectable temperature difference of the infrared thermography device [26]. Thermogram temperature thresholds used for optimal diagnostic accuracy varied from 0.1–3 °C between the included studies. These were derived through sequential imaging over a period for a smaller patient sample, which was then used to identify the different sensitivity and specificity via an AUROC curve. Martinez-Jimenez et al. [19] and Ganon et al. [18] have favored a high specificity over sensitivity, as it would be detrimental to subject superficial burn wounds to surgical management due to misdiagnosis because of the implied anesthetic risks, surgical complications, and costs. Furthermore, the heterogeneity secondary to non-standardized protocols and varying thermal sensitivity of the cameras used resulted in a wide range of varying sensitivities and specificities. However, Martinez-Jimenez et al. [19] suggested that the Glamorgan Protocol [27] enables repeatability and thus a standardized method of thermal testing.

Clinical assessment remains the most pervasive diagnostic method for determining burn depth, despite only 60% to 75% accuracy [1]. Clinical assessment varies in efficacy due to differences in experience and expertise levels of burn specialists and surgeons. The current gold standard of burn-depth assessment is tissue punch biopsy and subsequent histological investigation—though this suffers from the same factors of variance as it is dependent on subjective, not objective, human interpretation [28,29]. ICG is a possible alternative diagnostic method that has been shown to determine if a burn will heal within a 21-day timeframe [30]. ICG has the advantage of familiarity as fluorescent imaging is used in a variety of medical procedures and is supported by the literature [29]. ICG is minimally invasive, but it is not noninvasive. Additionally, allergic or pseudoallergic reactions to the contrast used in ICG are possible [31]. As an example, there have been serious cases of iodide mumps occurring in patients with no allergic history and no previous issues with iodine-based contrast [32]. LDI is noninvasive and has reported high accuracy in burn-depth diagnosis, which was 79.5% at Day 1 and 100% at Day 8 [20]. Having approval by the Food and Drug Administration in the United States, it is the most widely used noninvasive procedure for burn depth [11]. Its popularity lies in the fact that no routine second visit is required and has a standardized interpretation protocol unlike the IRT which requires evaporative cooling and the need for a controlled environment. However, LDI requires patients to remain still, which presents difficulty in the paediatric population or patients with unmanageable pain [11].

We propose IRT as a viable diagnostic tool as it combines the high-fidelity diagnostic power of ICG and LDI with a low-cost, noninvasive procedure. FLIR (Forward looking Infrared) ONE is a commercial product that turns a smartphone into thermal cameras that are small and portable and capable of detecting small temperature differences (0.05 °C) of large, affected areas in real-time [33]. They can be attached to phones and tablets, allowing for measurements to be taken in not only specialist burn centers but also in general practice and hospital emergency departments [33]. Moreover, because of the portability and relative ease of use, infrared thermography devices can find use in triaging where burn severity can be determined—separating burn wounds that would need specialist treatment from those that could heal with conservative treatment [27]. Issues with IRT are minimal, with the primary concern being successful base-line establishment. Carrière et al. [26] found the selection of appropriate reference areas of unaffected skin to be difficult as there were differences between extremities. Humidity, room temperature, evaporation, and wound exposure time are environmental factors that can affect measuring accurate temperatures [26]. However, the use of relative temperature differential (RTD) to compare a wound site with a healthy control area on the patient eliminates the significance of environmental temperature interference and individual body temperature variance [33].

One of the major limitations of our review is the lack of a standardized protocol for IRT, incorporating heterogeneity between the included studies. The need for evaporative cooling and a controlled external temperature is cumbersome and not always feasible in clinical practice. Moreover, the variation in temperature threshold stems from a lack

of precedence. IRT in the assessment of burn depth is a relatively novel concept, and most studies relied on sequential imaging to obtain temperature differences. Similarly, the effect of scan timing on outcome is still unclear. There is considerable heterogeneity in the thermal sensitivity of the devices used, timing of measurement, and temperature threshold used, thus resulting in a wide range of sensitivity and specificity. Five of the six included studies [17–19,23,25] utilized clinical outcomes as their reference standard, based on the assumption that burns which have not healed within 2–3 weeks need excision and grafting and are considered as deep dermal. The lack of histological results is a major disadvantage of the included studies. We have found that the diagnostic process of burn-depth analysis is typically bimodal—where a specialist determines if a burn will heal within 21 days or if the burn will require surgical intervention, leading to a lack of follow-up visits to further assess healing. Moreover, the current dominant method of clinical assessment provides little foundation for a hierarchical ranking of ICG, LDI, and IRT as all three indeed exceed accuracy in both sensitivity and specificity by wide margins. Finally, there are limitations with IRT as a technology as variance in the image does not provide an accurate absolute temperature. However, the applications of temperature as a diagnostic modality in burn-depth assessment relies on using RTD by comparing a healthy control area against the burn site [33].

## 5. Conclusions

Timeliness and accuracy of diagnoses are critical pillars of patient care. Cost and invasiveness are mitigating factors that must be considered alongside these. The authors conclude that IRT is a suitable, low-cost diagnostic tool for burn-depth assessment as it provides fast and accurate results without risk to the patient. Additional visits due to misdiagnosis, complications from reactions to contrast, specific timeframes for LDI, and all other factors considered—IRT as an adjunct to clinical assessment is a viable long-term solution. IRT excels in the areas of interest to burn patients, providers, stakeholders, and policy makers. The accessibility and reliability of IRT via cost of equipment, ease of the procedure and portability, noninvasiveness, lack of specific time frames to receive accurate results, and the high sensitivity and specificity of results greatly outweigh complications from environmental factors and the need for new training and equipment.

**Supplementary Materials:** The following supporting information can be downloaded at: https://www.mdpi.com/article/10.3390/ebj3030038/s1, File S1: Keywords and Search Strategy

**Author Contributions:** Conceptualization, D.M.R. and F.J.R.; methodology, D.M.R.; software, F.J.R.; validation, D.M.R., A.A. and C.P.; formal analysis, F.J.R.; data curation, D.M.R., A.A. and C.P.; writing—original draft preparation, D.M.R. and A.A.; writing—review and editing, F.J.R. and C.P.; visualization, F.J.R.; supervision, D.M.R. All authors have read and agreed to the published version of the manuscript.

**Funding:** This research received no external funding.

**Conflicts of Interest:** The authors declare no conflict of interest.

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
