# Peer review of "The Use of Infrared Thermography (IRT) in Burns Depth Assessment: A Diagnostic Accuracy Meta-Analysis"

_2673-1991, doi:10.3390/ebj3030038_

Round 1
Reviewer 1 Report
The review "The use of Infrared Thermography (IRT) in burns depth assessment: A diagnostic accuracy meta-analysis" is a very well written review.
The authors followed all reasonable guidelines to produce a meaningful review.
Introduction is clear, methodology is stated comprehensively.
Results and discussion is of good quality.
Author Response
Thank you for your comments, they are very much appreciated
Reviewer 2 Report
In this manuscript, the authors summarized and compared six related papers to study how infrared thermography (IRT) would work for burn depth assessment. Generally it is well written and can be accepted. There are several minor revisions needed, as follows.
- The full name and abbreviation “infrared thermography [IRT]” should be given at the very beginning. But it was given in section 2.2.4 on page 3.
- In the manuscript, the authors used both “thermography” and “infrared thermography”. According to my understanding, thermography is infrared thermography, is that correct?
- Page 3 section 2.2.4, the authors said that “The advantage of IRT is … as deeper burns will return to body temperature slower than superficial and superficial partial thickness burns.” Is it the advantage of IRT or specifically the activate IRT?
- The abbreviations “ICG” and “LSI” were given twice in both section 1 and section 2.2.5.
- Page 4, what is FLIR?
- Page 11, section 3.3, the authors talked about two studies (15, 18) trying to obtain a maximum specificity. Why is specificity more important than sensitivity? What was the sensitivity in reference 18 with a 100% specificity?
- Page 12, the two paragraphs before section 3.4 seem to be one part, as the first sentence in the second paragraph is still about reference 23.
- Page 12, inconsistent format for reference citations, i.e. (25) and [21].
- In the discussion section, the authors said that this is the first study to assess the accuracy of thermography in burn depth analysis. However, it seems to me that the section 3.4 was the main part that talked about burn depth. If so, please strengthen this part. Also, in the first sentence of section 4, it should be “burn depth” instead of “burns depth”.
Author Response
- The full name and abbreviation “infrared thermography [IRT]” should be given at the very beginning. But it was given in section 2.2.4 on page 3. – amended
- In the manuscript, the authors used both “thermography” and “infrared thermography”. According to my understanding, thermography is infrared thermography, is that correct? – Yes it is correct, the manuscript has been amended to replace all ‘thermography’ by ‘infrared thermography’ to avoid further confusion
- Page 3 section 2.2.4, the authors said that “The advantage of IRT is … as deeper burns will return to body temperature slower than superficial and superficial partial thickness burns.” Is it the advantage of IRT or specifically the activate IRT? – it is the advantage of IRT as it measures the blood flow in terms of temperature changes and hence can determine different depths of burn wounds
- The abbreviations “ICG” and “LSI” were given twice in both section 1 and section 2.2.5. - amended
- Page 4, what is FLIR? – forward looking infrared - changed to IRT to avoid confusion
- Page 11, section 3.3, the authors talked about two studies (15, 18) trying to obtain a maximum specificity. Why is specificity more important than sensitivity? What was the sensitivity in reference 18 with a 100% specificity? – The sensitivity in 18 was 56.7% due to the segmental testing, and specificity was prioritised when targeting thresholds as accurately ruling out deep burns enables conservative management for the appropriate cohort of patients, and reduces the operative burden and perioperative complications of burns excision and avoids unnecessary general anaesthesia in children – added to paragraph to clarify reason
- Page 12, the two paragraphs before section 3.4 seem to be one part, as the first sentence in the second paragraph is still about reference 23. - merged
- Page 12, inconsistent format for reference citations, i.e. (25) and [21]. - amended
- In the discussion section, the authors said that this is the first study to assess the accuracy of thermography in burn depth analysis. However, it seems to me that the section 3.4 was the main part that talked about burn depth. If so, please strengthen this part. – section 3.4 refers to our statistical results which are further elaborated and commented upon in the discussion section
- Also, in the first sentence of section 4, it should be “burn depth” instead of “burns depth”. - amended
Reviewer 3 Report
This is a very interesting review about IRT as means for burn depth assesment.
Consider revision of Table 1, here the column "types of burn" is given two times, and the meaning is not clear - I suggest removal.
Otherwise the methodology of all cited studies seems to lack thorough criteria to distinguish between superficial and deep burn (e.g. punch biopsies, as mentioned in the methods section) - this should be pointed out clearly.
Compliments go to the authors for their work.
Author Response
Consider revision of Table 1, here the column "types of burn" is given two times, and the meaning is not clear - I suggest removal. – removed duplication of columns
Otherwise the methodology of all cited studies seems to lack thorough criteria to distinguish between superficial and deep burn (e.g. punch biopsies, as mentioned in the methods section) - this should be pointed out clearly. – amended and clarified
Compliments go to the authors for their work. - Thank you so much, we really appreciate it
Reviewer 4 Report
Thank you for your manuscript. It is well-written. While multiple methods of depth assessment are somewhat better than expert clinical judgement, none are sufficiently better that they have gained wide traction in the market. Perhaps, as you say, a smart-phone thermal imaging app would be of enough use and sufficiently easy to use that clinicians would adopt it. Only widespread use would reveal real world utility.
Author Response
Thank you for your comment, we appreciate it very much